

# Insecticides may facilitate the escape of weeds from biological control

Elizabeth K. Rowen[1,2], Kirsten Ann Pearsons[1], Richard G. Smith[3], Kyle Wickings[4] and John F. Tooker[1]

[1] Department of Entomology, The Pennsylvania State University, University Park, PA, United States of America
[2] Entomology, University of California, Riverside, Riverside, CA, United States of America
[3] Department of Natural Resources and Environment, University of New Hampshire, Durham, NH, United States of America
[4] Department of Entomology, Cornell AgriTech, Cornell University, Geneva, NY, United States of America

## ABSTRACT

**Background**. Preventative pesticide seed treatments (hereafter preventative pest management or PPM) are common corn and soybean treatments, and often include both fungicides and neonicotinoid insecticides. While PPM is intended to protect crops from soil-borne pathogens and early season insect pests, these seed treatments may have detrimental effects on biological control of weed seeds by insects.

**Methods**. Here, in two 3-year corn-soy rotations in Pennsylvania USA, we investigated a PPM approach to insect management compared to an integrated pest management approach (IPM) and a "no (insect) pest management" (NPM) control. This was crossed with a grass cover crop to see if this conservation practice can help recover the ecosystem services affected by chemical pest management practices. We hypothesized that PPM and IPM approaches would release weed seeds from biological control by insects but cover crops would increase biological control. We measured the effect of these treatments on the weed-seed bank, mid-season weed biomass, granivorous insect activity-density, and weed-seed predation.

**Results**. We found that, contrary to our hypothesis, planting a cover crop decreased carabid activity-density without consistent differences in weed-seed predation. Pest management and cover crop treatments also had inconsistent effects on the weed-seed bank and mid-season weed biomass, but insecticide use without a cover crop increased the biomass of likely glyphosate-resistant marestail (*Erigeron canadensis* L.) at the end of the trial. Our results suggest that reducing insecticide use may be important when combating herbicide-resistant weeds. We found planting cover crops and/or avoiding the use of insecticides may combat these problematic weeds.

# INTRODUCTION

In the US, approximately 90% of corn and >50% of soybean seeds are coated with pesticidal seed treatments that typically include a mixture of fungicides and a neonicotinoid insecticide (*Douglas & Tooker, 2015*). These prophylactic pesticides are meant to protect seeds and

Corresponding author
Elizabeth K. Rowen,
elizabeth.rowen@ucr.edu,
elizabethrowen@gmail.com

young plants from a suite of fungal pathogens and early-season insect pests like seed corn maggot, wireworms, cutworms, and white grubs (*Douglas & Tooker, 2015*). Agricultural chemical companies market their products to growers emphasizing that the pesticides provide a sort of insurance policy against difficult-to-scout early season pests (*Hurley & Mitchell, 2017*). It is important to recognize, however, that use these seed treatments may not be needed to control pests, particularly in cooler climates (*Mourtzinis et al., 2019*: *Labrie et al., 2020*; *Smith, Baute & Schaafsma, 2020*). In contrast to more northern latitudes, in the Southern United States, where early season pests are abundant and diverse, estimates suggest that pesticide seed treatments can provide economic advantages to soybean and corn production (an average increase of $31–$50/ha for soy and corn; (*North et al., 2016*; *North et al., 2018*). Whether warranted or not, many farmers do not know that these pesticides are on their seeds (*Hitaj et al., 2020*), nor have options to choose the seed treatments that best suit their pest pressure and management needs.

Despite providing advantages in some situations, recent research on seed treatments has cast doubt on their widespread benefits for pest control and yield protection, especially their insecticidal constituents (*Labrie et al., 2020*; *Smith, Baute & Schaafsma, 2020*). Meanwhile, other research has revealed that the nitroguanidine neonicotinoids (*i.e.,* clothianidin, thiamethoxam, and imidacloprid) coated on seeds are water soluble, and readily leach from fields and enter aquatic systems where they can have significant non-target effects (*e.g., Hladik, Kolpin & Kuivila, 2014*; *Hladik & Kolpin, 2015*; *Miles et al., 2017*; *Frame et al., 2021*). While many non-target effects of neonicotinoid-containing seed treatments are external to farms, neonicotinoid insecticides in field soil, prey, and plant tissues can also pose risks to insects providing in-field ecosystem services. Much of the prior work on in-field effects of neonicotinoids has focused on pollinators (*Main et al., 2021*; *Pecenka et al., 2021*; *Ward et al., 2023*) and invertebrate predators such as carabid beetles, which are particularly effective at controlling otherwise difficult-to-control pests like slugs but are sensitive to neonicotinoids applied as seed treatments (*Douglas, Rohr & Tooker, 2015*; *Penn & Dale, 2017*; *Rowen et al., 2022*; *Mugala et al., 2023*). Neonicotinoid seed treatments have been found to decrease in-field populations of pest-regulating natural enemies (*Douglas & Tooker, 2016*), with at least one case of relaxed predation resulting in lower crop yield from non-insect pests (*Douglas, Rohr & Tooker, 2015*).

In addition to being important predators of invertebrate pests, carabids and other epigeal predators such as ants can also be effective weed-seed predators (*Baraibar et al., 2009*; *Sarabi, 2019*; *Kulkarni et al., 2017*). Carabids feeding on seeds, for example, can reduce weed biomass as much as 81% following seed rain (*Blubaugh & Kaplan, 2016*), but insecticides can interfere with these benefits. For instance, relative to control plots, predation of lambsquarters (*Chenopodium album* L.) seeds decreased in corn plots that received an application of a pyrethroid insecticide that killed weed-seed predators (*DiTommaso et al., 2014*). Coupled with the broad-spectrum fungicides used in seed treatments, which can inadvertently protect weed seeds from fungal attack (*Mohler et al., 2012*; *Gomez, Liebman & Munkvold, 2014*; *Smith et al., 2016*), pesticidal seed treatments may indirectly increase abundance of weed seeds in seed banks by releasing seeds from biological control. Although well-timed herbicide applications can keep most weeds at

bay, releasing weed seeds in the soil seed bank from biological control may exacerbate management challenges related to herbicide-resistant weeds. As a result, biological control of weed seeds is an important component of integrated weed management and resistance management that may be disrupted by pesticide use (*Harker & O'Donovan, 2013*).

An alternative to preventative pest management including seed treatments is integrated pest management (IPM; *Stern et al., 1959*). In agricultural systems managed with IPM, pesticides are applied only if pest populations exceed economic thresholds. Such pesticide applications are used as last resort after other control methods fail to control pest populations. Consequently, IPM is often less ecologically disruptive compared to insurance- or calendar-based use of pesticides (*Stenberg, 2017*). However, the primary targets of seed treatments, early-season soil-borne insect pests, can be challenging to manage using IPM because subterranean pests are hard to control through rescue treatments after crops are planted. Therefore, farmers may appear to face a tradeoff: commit to IPM, including avoiding seed treatments, and potentially leave young crop plants vulnerable to damage from unseen insect or fungal pests, or preventatively deploy seed treatments that may be able to control early season pest damage and accept their non-target effects.

A potential solution to this apparent tradeoff is to use conservation-based farming practices to build natural-enemy populations that can decrease the need for pesticides. While use of pesticidal seed treatments is increasing in the US (*Douglas & Tooker, 2015*; *Douglas et al., 2020*), adoption of conservation-based agricultural practices is also growing. No-till farming is standard in many parts of the northern Corn Belt of the U.S. and is becoming more common elsewhere (*National Agricultural Statistics Service, 2020*), while adoption of cover crops, particularly in northeastern and Mid-Atlantic states, is growing because of benefits to weed management and soil quality (*Wallander et al., 2021*). Winter cover crops that produce significant biomass by spring can out-compete annual weeds that germinate in the weeks prior to planting (*Teasdale, 1996*). Further, after they are terminated, cover-crop residues on soil surfaces in no-till systems can provide further control of weeds (*Schipanski et al., 2014*; *Daryanto et al., 2018*). In addition to these benefits, cover crops can provide overwintering habitat, and their decomposing residue supports the detritivores and the brown food web and increases predator populations (*Halaj & Wise, 2002*). Because weed-seed predators are often omnivorous, cover crops can help stabilize and support their populations (*Blubaugh et al., 2016*). However, increasing use of neonicotinoid seed treatments may counteract some of the benefits of cover crops, including reducing the potential of weed-seed predators to contribute to biological control of weeds.

To explore the potentially competing influences of seed-applied pesticides and cover crops on weed management, we conducted a three-year field experiment to address the following questions: (1) How do preventative pest management (PPM), including pesticidal seed treatments, and IPM interact with cover crops to influence communities of weed-seed predators and weed-seed predation? (2) How do PPM, IPM, and cover crops affect the weed-seed bank? (3) How do PPM, IPM, and cover crops influence weed composition and weed biomass in the field?

We expected that insecticide use *via* PPM or IPM would reduce the abundance and diversity of weed-seed predators, particularly carabids, resulting in less weed-seed

 

predation. By disrupting weed-seed predation, we expected insecticides would indirectly increase overall abundance of weeds in the weed-seed bank, and possibly decrease the diversity of those weeds, and this effect would be stronger in PPM where fungicides may also decrease fungal biological control of weed seeds. Cover crops, however, may bolster natural enemy communities and their function, resulting in greater predation of weed seeds. Even in the context of standard herbicide use, we expected cover crops to help suppress weed biomass by reducing weed emergence and seed production (*Fernando & Shrestha, 2023*). Lastly, we expected that all these effects should become more pronounced over time as effects of seed treatments and cover crops accumulate.

# MATERIALS & METHODS

## Field sites

We established our three-year field experiment in two 1.5-ha fields at the Penn State Russell E. Larson Agricultural Research Center (Rocksprings, PA, 40°42′42″N, 77°57′51″W), and identified them as "North" and "South" fields, referring to their location relative to a main road that bisects the research farm. In 2016, the year before the experiment began, five of six blocks in the South field grew soybeans, and one block grew a combination of sunflower mixed with other harvestable forage crops. In the North field, three blocks grew wheat, and three blocks grew soybeans. In spring 2017, we established a factorial field experiment to quantify the interactive effects of pesticide seed treatments and grass cover crops. The experiment was established as a soy-corn-soy rotation on the North field and as a corn-soy-corn rotation on the South field. We divided each field into 12.2 × 33.5 m plots, laid out in a randomized complete block design (RCBD) with six treatments (three levels of pest management by two levels of cover cropping) each replicated six times in each field.

The pest management treatments consisted of a "preventative" pest management treatment ("PPM"), an integrated pest management treatment (IPM), and a control ("no pest management": "NPM"). In the PPM treatment, seeds were treated with a neonicotinoid insecticide and mixture of fungicides prior to planting (*Rowen et al., 2022*, Table S1). In the IPM treatment, we scouted plots and treated fields if insect pest populations exceeded economic thresholds. During the three-year experiment, the IPM plots received a single insecticide application: an in-furrow application of tefluthrin (Force 3G, 5 kg ha$^{-1}$ [Syngenta]) at planting in 2018 to control white grubs (Scarabaeidae; mostly Japanese beetles *Popillia japonica* Newman). Thus, in 2017, pest management practices in the IPM and NPM treatments were identical, while in 2019, IPM and NPM treatments were both untreated but had different treatment histories.

Cover-crop treatments included oats (*Avena sativa* L.) planted in spring of 2017 and 2019, and cereal rye (*Secale cereale* L.) planted in the fall of 2017. We used spring-established oats when planting conditions in fall were not ideal for establishing cereal rye. One to two weeks before planting corn or soybeans, we applied herbicides (RoundUp PowerMax or a combination of Impact, Accent, Banvil and DegreeExtra; Table S1) to all plots to terminate cover crops and manage early-season weeds across both fields. The terminated cover crops

**Table 1  Weed biomass and abundance metrics (mean ± SE) across sampling dates.**

| Sampling event | Metric | 2017 | | 2018 | | 2019 | |
|---|---|---|---|---|---|---|---|
| | | North | South | North | South | North | South |
| Pre-plant biomass | Sample collection date | 17 May | 12 May | 8 May | 14 May | 15 May | 7 May |
| | Weeds (g) | 8.6 ± 2.5 | 30.3 ± 4.9 | 7.8 ± 1.7 | 28.6 ± 3.7 | 15.4 ± 1.8 | 11.6 ± 1.9 |
| | Total (weeds + cover crops, g) | 15.2 ± 2.4 | 34.2 ± 4.9 | 12.8 ± 1.9 | 34.7 ± 3.7 | 26.5 ± 1.8 | 18 ± 1.8 |
| Pre-plant weed seed bank | Soil collection date | 15 May | 8 May | 9 May | 16 May | 15 May | 6 May |
| | Forb abundance (# seedlings) | 60.1 ± 6.6 | 57.4 ± 4.6 | 39.2 ± 3.9 | 31.8 ± 4.2 | 17 ± 2.8 | 2.9 ± 0.5 |
| | Grass abundance (# seedlings) | 3.2 ± 0.5 | 66.5 ± 8.6 | 1.7 ± 0.5 | 15.8 ± 2.6 | 0.2 ± 0.1 | 3.2 ± 0.7 |
| | Richness | 6.4 ± 0.4 | 10.9 ± 0.5 | 6.3 ± 0.4 | 7.5 ± 0.6 | 3.6 ± 0.3 | 2.3 ± 0.3 |
| Mid-season biomass | Sample collection dates | 24 Aug | 15 Aug | 15 Aug | 29 Aug –5 Sep | 21 Aug –22 Aug | 13 Aug –15 Aug |
| | Forb abundance (g m$^{-2}$) | 3.5 ± 0.7 | 13.7 ± 3.0 | 10.9 ± 2.7 | 25.0 ± 4.7 | 9.1 ± 2.0 | 10.3 ± 3.1 |
| | Grass abundance (g m$^{-2}$) | 2.1 ± 0.9 | 50.2 ± 6.0 | 0.2 ± 0.1 | 123.4 ± 11.9 | 115.4 ± 9.8 | 5.4 ± 1.4 |
| | Forb richness | 3.1 ± 0.2 | 2.1 ± 0.3 | 3.2 ± 0.3 | 6.0 ± 0.3 | 5.7 ± 0.3 | 2.8 ± 0.2 |

were not removed. In 2017 and 2018, we applied herbicides a second time (RoundUp and Accent) at the end of June or early July (Table S1; also in *Rowen et al., 2022*).

## Spring weed and cover crop biomass

We harvested weed and cover-crop biomass two to three weeks before planting (Table 1). We collected all above-ground plant biomass from three, randomly spaced 0.25-m$^2$ quadrats in each plot. Cover-crop biomass was collected and handled separately from weed biomass. We dried all biomass samples in a 60 °C drying oven for at least 5 days before weighing.

## Weed-seed bank

We used direct-germination assays to assess changes to the weed-seed bank in response to our treatments. We sampled the germinable weed-seed bank in early May each year (Table 1), before terminating weeds and cover crops or planting corn or soy. For each plot, we pooled nine, evenly spaced soil samples that were collected using a bulb planter (five cm wide, 10 cm depth; Yard Butler, San Diego, CA, USA). We sieved (1-cm mesh) and homogenized these pooled samples, then transferred them to paper bags to air-dry in a greenhouse for at least five days.

We subsampled 946 cm$^3$ of air-dried soil from each sample, which we spread across an equal amount of soilless potting media (Promix) in standard plastic planting trays (28 × 54 × 6 cm). After an initial watering, we watered these trays as needed (2–3x per week) depending on ambient temperature and weed emergence. We maintained assays for six months (July to January), regularly identifying, counting, then removing seedlings as they emerged. Species that were challenging to identify as seedlings were transplanted to pots to continue growing until they could be accurately identified.

## Weed biomass in August

To understand the influence of our treatments on weed communities, we assessed weed biomass in August of each year (Table 1). When assessment could not be completed in a single day, we assessed weeds by block. For each plot, we identified and collected all

**Table 2  Mean predator abundance and activity for each sampling date.** Predator trap capture (means ± SE) and weed seed predation across sampling dates.

| | Metric | 2017 | | 2018 | | 2019 | |
| --- | --- | --- | --- | --- | --- | --- | --- |
| | | North | South | North | South | North | South |
| Planting | | 2 Jun | 19 May | 30 May | 14 June (Blocks 3-6), 26 June (Blocks 1-2) | 22 May | 17 May |
| Pitfall traps | Sample collection (Early) | 23 Jun | 9 Jun | 15 Jun | 6 Jul | 31 May | 7 Jun |
| | Carabid activity-density[a] | 1.3 ± 0.3 | 1.1 ± 0.3 | 0.4 ± 0.1 | 2.9 ± 0.3 | 0.5 ± 0.2 | 0.6 ± 0.2 |
| | Ant activity-density | 21.4 ± 1.9 | 9.6 ± 1.1 | 19.4 ± 3.3 | 16.2 ± 1.6 | 6.9 ± 0.9 | 23.9 ± 3.7 |
| | Sample collection (Late) | 19 Sept | 1 Sept | 10 Aug | 31 Aug | 23 Aug | 16 Aug |
| | Carabid activity-density | 5.9 ± 1.4 | 16.5 ± 3.0 | 2.3 ± 0.5 | 14.4 ± 2.6 | 2.2 ± 0.3 | 0.1 ± 0.1 |
| | Ant activity-density | 5.9 ± 0.9 | 2.4 ± 0.5 | 3.7 ± 0.5 | 5.6 ± 1.7 | 4.7 ± 1.1 | 14.7 ± 2.0 |
| Sentinel seeds | Sample collection (Early) | 27 Jun | 12 Jun | 18 Jun | 9 Jul | 3 Jun | 10 Jun |
| | Total seed predation[b] | 5.2 ± 0.8 | 3.2 ± 0.6 | 14.2 ± 0.7 | 12.0 ± 1.0 | 8.2 ± 0.6 | 3.1 ± 0.5 |
| | Sample collection (Mid) | 8 Aug | 24 Jul | 17 Jul | 6 Aug | 23 Jul | 15 Jul |
| | Total seed predation | 6.3 ± 1.0 | 12.1 ± 1.4 | 14.5 ± 1.3 | 34.6 ± 1.5 | 7.3 ± 0.7 | 15.81 ± 1.4 |
| | Sample collection (Late) | 10 Sept | 7 Sept | 13 Aug | 3 Sept | 26 Aug | 19 Aug |
| | Total seed predation | 2.5 ± 0.4 | 27.0 ± 1.8 | 35.2 ± 1.5 | 37.5 ± 1.8 | 17.5 ± 1.2 | 23.0 ± 1.9 |

**Notes.**
[a]Activity-density measured per pitfall trap.
[b]Total seed predation of both pigweed and foxtail combined (total possible seeds = 50).

aboveground weed biomass from three, randomly spaced 0.25-m² quadrats. In 2017, we harvested plants en-masse and brought them into to the lab for identification; for forbs, we identified plants to species, whereas we grouped all grasses together. In 2018 and 2019, we identified all forb and grass weeds to species in the field and collected each species into separate paper bags. Some plots had such small amounts of a particular species that we noted its presence but did not harvest it for collection. We dried harvested biomass for at least 5 days at 55 °C and weighed each species.

## Weed-seed predator community

To understand the influence of our treatments on seed-eating invertebrates, we assessed weed-seed-predator communities twice a year, in late June/early July and late August/early September. We defined the weed-seed-predator community as granivorous carabid beetles and ants, which we captured using two, evenly spaced pitfall traps per plot. Pitfall traps were constructed from 946 mL cups installed flush in the ground so that ground-dwelling insects would pass over them and fall in. We used a 50:50 mix of propylene glycol and water (∼60 mL per trap) as a killing agent. When traps were in-use, we placed a 20-cm plastic plate (propped ∼5 cm above the soil surface using nails) over them to protect traps from rain. We left traps open for 72 hr in June/July and Aug/Sept (Table 2). We returned trapped specimens to the lab, rinsed them with water, and transferred them to ethanol (70%) for storage and identification. When not in-use, we attached tight-fitting lids to the pitfall traps to avoid capturing insects between sampling events. In 2019 in the South field, vertebrate pests, likely raccoons, destroyed the majority of our traps during both sampling events, so we were unable to include those 2019 South field data in our analysis.
We identified carabid beetles to species using keys in *Bousquet (2010)* and all other insects to order. Here, we report on species that are classified as predominantly weed-seed predators, including ants (Formicidae) and carabid beetles in the genera *Anisodactylus*, *Amara*, *Harpalus*, *Notiobia*, and *Bembidion* (*Larochelle, 1990*; *Lundgren, 2009*). Taxa that predominantly feed on insects and other invertebrates are reported and discussed in *Rowen et al. (2022)*.

## Weed-seed predation

We deployed sentinel seeds to measure weed seed predation by invertebrates in the field. In 2017, we deployed sentinel seed cards comprising 30 seeds of red-root pigweed (*Amaranthus retroflexus* L.) and 20 seeds giant foxtail (*Setaria faberi* L.) that we glued to a 4 × 9 cm piece of 60 grit sandpaper (*Westerman et al., 2003*). Because this method proved delicate and difficult to transport, we switched to a different deployment method in 2018 and 2019 and removed instances where we recovered more than the number of seeds we put out from the dataset. In 2018 and 2019, each trap comprised double-sided tape (Duck® Brand Indoor Heavy Traffic Carpet Tape) attached to the bottom of an inverted Petri dish (5-cm diameter) with 30 pigweed and 20 giant foxtail seeds scattered across the surface. We then adhered sifted sand (Quickrete Play Sand) across the remaining sticky areas of the tape (*Gallandt, 2005*; *Ward et al., 2011*) to prevent predators from getting stuck to the dishes. We installed seed cards or seed dishes in fields, with each placed inside vertebrate exclusion cages made from hardware cloth (10 cm wide × 8 cm tall, with 1-cm mesh with a plastic lid). We deployed three weed-seed cards (2017) or dishes (2018 & 2019) per plot and left them in place for 48 hr, three times each summer (Table 2). After 48 h in the field, we collected seed cards into envelopes (2017) and dishes into plastic bags (2018 & 2019) and brought them back to the lab to count the remaining number of whole pigweed and foxtail seeds.

## Statistical analyses

In general, we tested the interactions of pest management (PM), presence of a cover crop (CC), field (N: North, S: South) and year (2017, 2018, 2019) using generalized linear mixed models (GLMMs) with plot, and both directions of blocking from the Latin-square design as random intercepts using appropriate error distributions. For carabid and ant activity density, we included the interaction of month (June/July and Aug/Sept) in the model. Similarly, we included an interaction of month with all other categorical variables in modeling weed-seed predation (June, July, or August/September). For each model, we calculated estimated marginal means based on the full interaction model and plotted those means with 95% confidence intervals. To determine the relative effects of different factors, we then reduced the interactions in the models until they had the lowest AIC values. We never dropped a term from the model completely (*e.g.*, even if they were not significant, we included field or cover crop in the model). We ran all analyses, except weed-seed predation and forb biomass, using a Poisson or a negative binomial error distribution. Weed-seed predation models used a binomial distribution where the fate of each seed was calculated separately, and each seed card/dish, as well as the plot, was included as a random

intercept to account for non-independence. For forb biomass, we used a zero-inflated gamma error distribution with log link function. We conducted all GLMM analyses using the package 'glmmTMB' (*Brooks et al., 2017*) in R (v. 4.1.2; *R Core Team, 2018*). We used the package 'emmeans' to calculate estimated marginal means and to conduct pairwise post-hoc tests using a Tukey multiplicity adjustment (*Lenth, 2019*). To examine violations of homogeneity of variance and test the fit of our chosen distributions, we used the package 'DHARMa' (*Hartig, 2018*). To test for spatial autocorrelation of *E. candensis* in 2019, we used Moran's I test of Spatial Autocorrelation using the package 'ape' (*Paradis, Claude & Strimmer, 2004*).

## RESULTS

### Weed biomass in May/June

Weed biomass measured before cover crops were terminated and cash crops were planted differed among years and between fields (Field × Year: $\chi^2 = 36$, $df = 2$, $P < 0.0001$; Fig. 1, Table 1). As expected, weed biomass was often lower in plots where cover crops were planted (CC: $\chi^2 = 6.0$, $df = 1$, $P = 0.01$), though the magnitude of difference varied depending on the field (Field x CC: $\chi^2 = 4.3$, $df = 1$, $P = 0.04$). In the North field in 2017 and 2019, we observed reductions in weed biomass due to cover crops (2017: $P = 0.015$, 2019: $P = 0.002$). Pest management treatments had no effect on weed biomass before cash crop planting ($\chi^2 = 2.8$, $df = 2$, $P = 0.25$).

When including cover-crop biomass, total plant biomass in spring was consistently higher in cover-cropped plots compared to non-cover-cropped plots (CC: $\chi^2 = 29.3$, $df = 1$, $P < 0.0001$), although the effect of cover on total biomass depended on the field and year (Field × CC: $\chi^2 = 7.7$, $df = 1$, $P = 0.005$; Year × CC: $\chi^2 = 12.0$, $df = 2$, $P = 0.003$; Fig. S1).

### Weed-seed bank

The weed-seed bank was dominated by forbs from the North field and grasses in the South field (Fig. S2). Further, we saw different communities in different blocks in the North and South fields that corresponded strongly with previous field history (between blocks 3 & 4 in the North and block 6 from the other five blocks in the South).

We predicted that weed-seed predators and the fungicide in the pesticidal seed treatment would act on the weed-seed bank to alter the weed community. The forb weed-seed bank varied between fields and among years (Field × Year: $\chi^2 = 28.1$, $df = 2$, $P < 0.0001$, Fig. 2A, Table 1), and pest-management treatment decreased forb abundance (PM: $\chi^2 = 6.4$, $df = 2$, $P = 0.04$). We did not detect an interaction between pest management and field or year, but we did observe that the IPM treatment in 2017 and 2018 were lower than the other pest management treatments (Fig. 2A). Because the IPM treatment was not implemented until after the weed seed bank was sampled in 2018, this indicates that despite our Latin-square design, the abundance of seeds in the weed-seed bank was significantly different across the different pest-management treatments at the start of the project and was not attributable the treatments we imposed. As there do not appear to be consistent differences in plots year-to-year (*i.e.,* plots with high weed abundances in 2017 do not necessarily have high

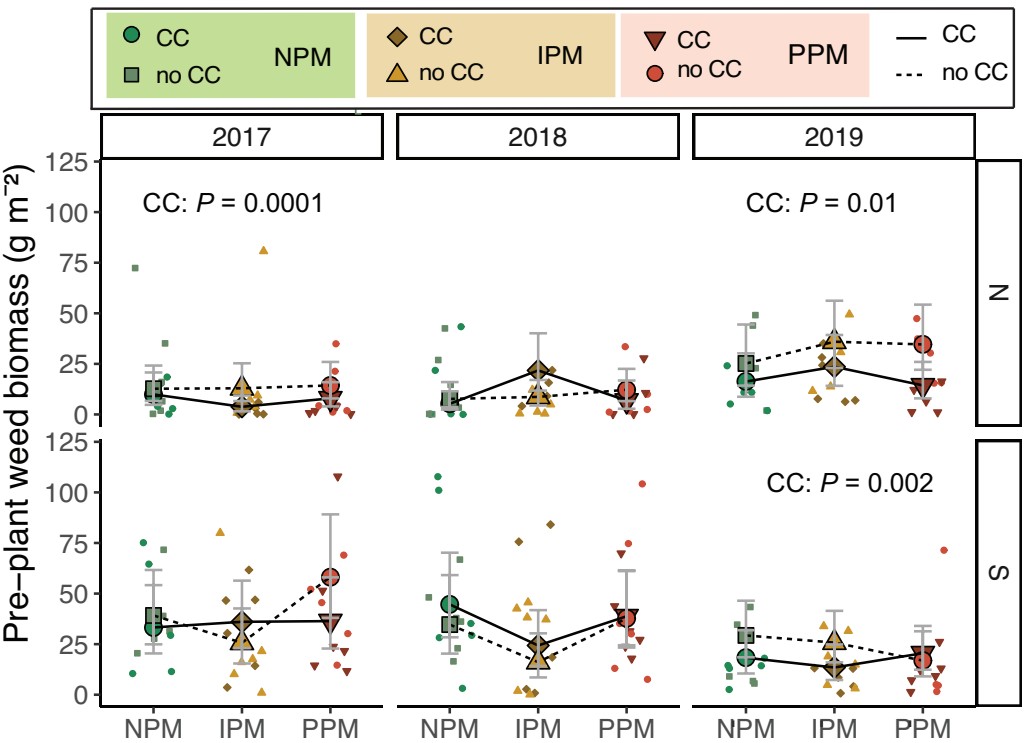

**Figure 1** **Weed biomass before planting.** Estimated marginal means (±95% confidence intervals (CIs)) of weed biomass (g m$^2$) before planting in each field and year. Significance of cover crop (CC) treatments included in panels where GLMM indicated cover crops had a significant effect by year/field slicing. Model information: GLM (lognormal distribution), $X^2 = 83.05$, $df = 13$, $P < 0.001$, $N = 204$. Means for treatments with cover crops indicated with a solid line, means without cover crop indicated by a dashed line. Raw data are shown as open small shapes behind means and CIs. We removed one outlier for graphing: in the South Field in 2017 (Plot 604S) with treatment PPM without cover crop, the weed biomass was 199 g m$^2$. North fields (N) on top panels, South fields (S) on bottom panels for each year.

abundances in 2018), we re-analyzed our data without 2017 in the model. When we looked at 2018 and 2019, we found that cover crops, rather than pest management treatment, decreased forb abundance in the weed-seed bank by 20% ($\chi^2 = 5.02$, $df = 1$, $P = 0.025$; Fig. 2B).

Similarly for grass seeds, we found that, again, the seed bank varied between fields and among years (Field × Year $\chi^2 = 9.4$, $df = 2$, $P = 0.009$, Fig. 3A). We also found that grass weed seed abundance was 25% higher when a cover crop was planted ($\chi^2 = 5.3$ $df = 1$, $P = 0.02$), although this appears to be again largely driven by differences in 2017. When 2017 was removed from the model, cover crops no longer affected the weed seed bank ($\chi^2 = 0.86$, $df = 1$, $P = 0.77$; Fig. 3B).

Because we expected weed seeds of different species would not be equally susceptible to pathogens, predators, or the suppressive effect of cover crops, we analyzed the effect of pest-management treatment and cover crops on species richness in the weed-seed bank. We found that species richness varied across fields and years (Field × Year: $\chi^2 = 43.5$, $df = 2$, $P < 0.0001$ Fig. S3A). We again found that species richness varied among pest-management

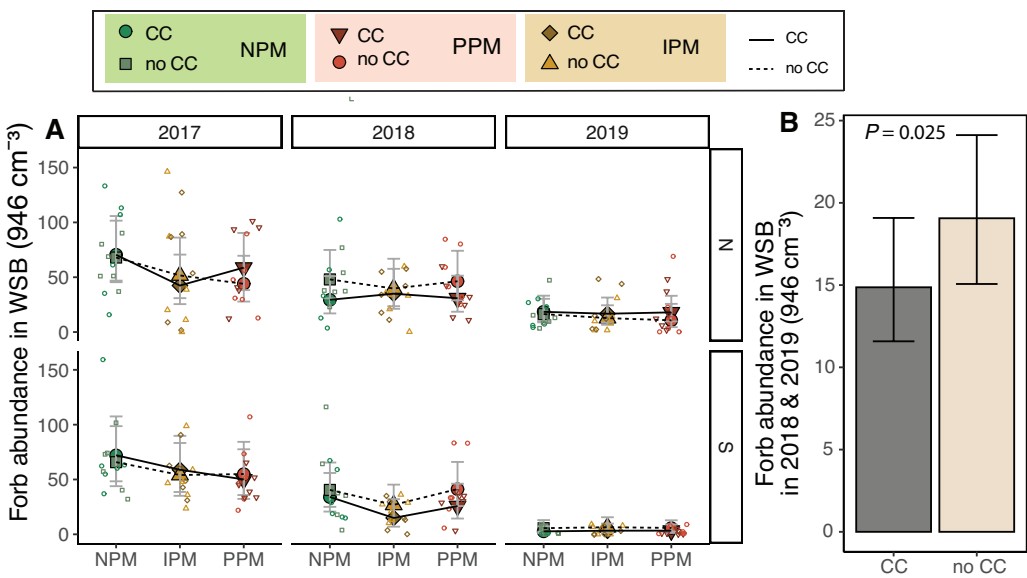

**Figure 2** **Forb abundance in the weed seed bank.** (A) Estimated marginal means (±95% CIs) of forb abundance (per 946 cm³ soil) in the weed-seed bank before planting in each field and year, and B) for 2018 and 2019 and fields combined. Model information: GLMM (Poisson distribution), $X^2 = 217$, $df = 8$, $P < 0.001$, $N = 216$. Means for treatments with cover crops indicated with a solid line, means without cover crop indicated by a dashed line. Raw data shown as open small shapes behind means and CIs.

treatments and was consistently lower in the IPM treatments compared to the PPM or NPM treatments, due to differences in 2017 and 2018, particularly in the North field (PM: $\chi^2 = 10.8$, $df = 2$, $P = 0.005$, Fig. S3B). Thus, as with abundance of weed seeds, the richness of the weed-seed bank started with different communities in the IPM treatments compared to other treatments despite our Latin-square design.

## Weed biomass in August

Because weeds compete with cash crops throughout the growing season, we measured weed biomass in August. When we examined the community composition of weeds in August, we saw patterns in weed communities depending on year and field. In the North field, the weed community had high abundances of dandelion (*Taraxacum* sp.), yellow woodsorrel (*Oxalis stricta* L.), and *C. album* in 2017, which was gradually replaced by marestail (*Erigeron canadensis* L.) and grasses by 2019. The South field had more grasses than other weeds in 2017 and 2018, and in 2019, dandelion was relatively common compared to grass (Fig. S4).

Examining the forb community, we found that mid-season forb biomass varied across field and years (Field × Year: $\chi^2 = 15.8$, $df = 2$, $P = 0.003$; Fig. S5, Table 1), but contrary to our hypothesis, did not respond to pest-management treatment ($\chi^2 = 1.3$, $df = 2$, $P = 0.5$) nor cover crop ($\chi^2 = 1.9$, $df = 1$, $P = 0.17$). One weed species *E. canadensis*, stood out as an increasing problem in the North field during the experiment (Fig. 4). In 2017, in the North field, we did not collect any *E. canadensis* in our samples. In the North field in 2018, we collected an average of 8.3 g *E. canadensis* m⁻² (95% CI [4.4–15.7]) and by 2019, we found an average of 93.4 g m⁻² (95% CI [61.4–142]). *Erigeron canadensis* in our trial did not

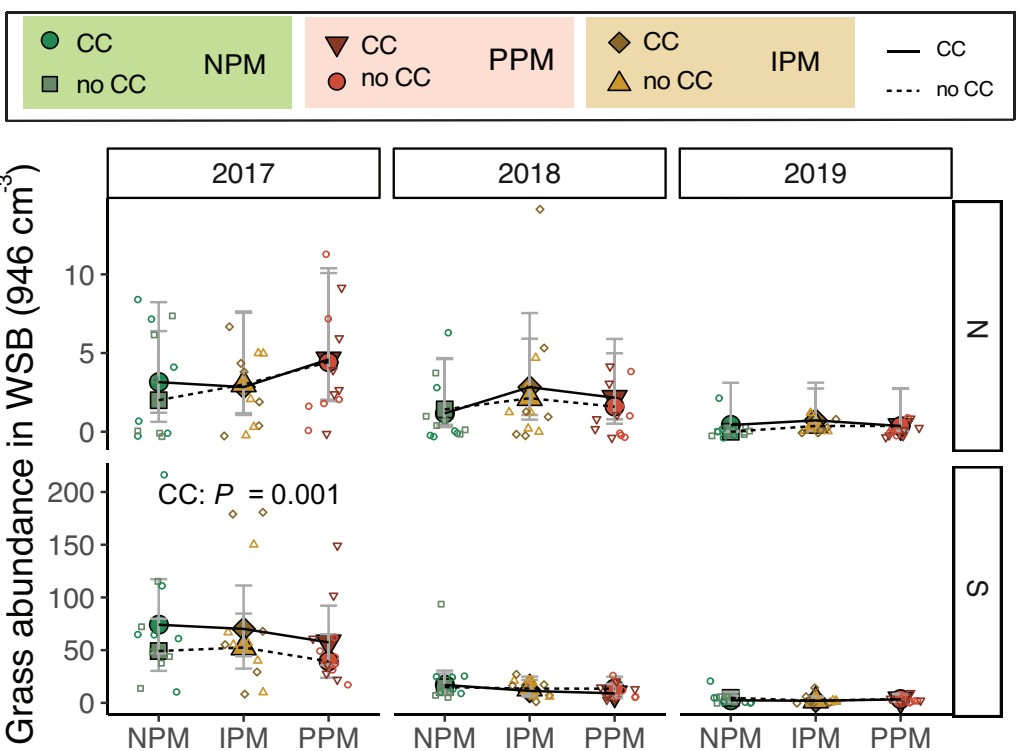

**Figure 3** **Grass abundance in the weed seed bank.** Estimated marginal means (±95% CIs) of grass abundance (per 946 cm³ soil) in the weed-seed bank before planting for each field and year. Means for treatments with cover crops indicated with a solid line, means without cover crop indicated by a dashed line. Model information: most parsimonious GLMM (Poisson distribution), $X^2 = 212$, $df = 8$, $P < 0.001$, $N = 216$. Significance of cover crop (CC) treatments included in panels where GLMM indicated cover crops had a significant effect by year/field slicing (using GLMM model with all interactions included: $X^2 = 233$, $df = 35$, $P < 0.001$, $N = 216$). Raw data shown as open small shapes behind means and CIs.

respond to glyphosate application in 2019. In 2019, treatments without a cover crop that used an insecticide (IPM or PPM) had significantly higher *E. canadensis* biomass than the other treatments (CC × PM: $\chi^2 = 6.9$, $df = 2$, $P = 0.03$, Fig. 4). Because *E. canadensis* seeds are wind-distributed, we tested for spatial autocorrelation within *E. canadensis* biomass in these plots. We found no evidence of autocorrelation (Moran's $I = 0.016$, $P = 0.18$).

Grasses were an important part of the weed community in the South field in 2017 and 2018, and in the North field in 2019. We analyzed each field year separately because there was almost no grass in the North field in 2018. In 2017, in both the North and South fields, planting a cover crop suppressed biomass of grass weeds into August (North: $\chi^2 = 4.8$, $df = 1$, $P = 0.03$; South: $\chi^2 = 48.9$, $df = 1$, $P < 0.0001$, Fig. 5, Table 1). In addition, in the South field in 2017, we found grass weed biomass was higher in the IPM plots, likely due to initial differences in the weed-seed bank (PM: $\chi^2 = 9.9$, $df = 2$, $P = 0.007$). In 2019 in the North field, we again found that cover crops marginally suppressed grass (CC: $\chi^2 = 3.5$, $df = 1$, $P = 0.06$). In the South field in 2018 and 2019, however, planting a cover crop increased grass biomass in August (2018: $\chi^2 = 8.7$, $df = 1$, $P = 0.003$; 2019: $\chi^2 = 3.4$, $df = 1$, $P = 0.07$). In 2018, the magnitude of this effect depended on the pest

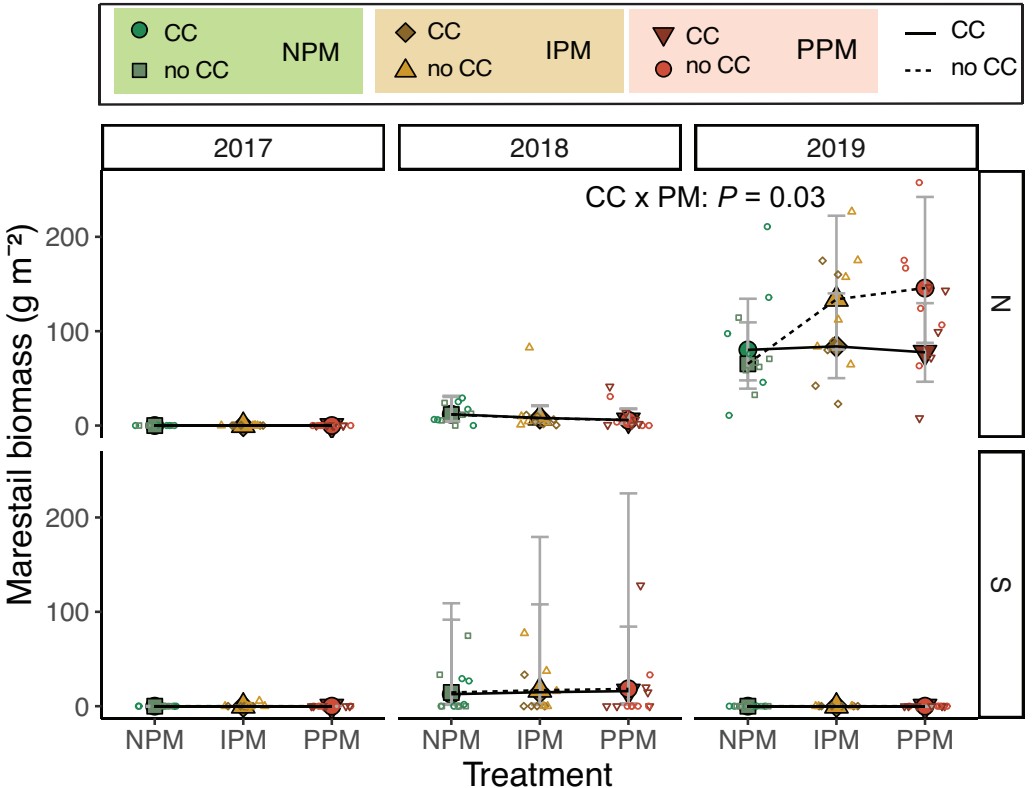

**Figure 4  Erigeron canadensis biomass in August.** Estimated marginal means (±95% CIs) of *E. canaden-sis* biomass (g m$^{-2}$) in August for in each field and year. Means for treatments with cover crops indicated with a solid line, means without cover crop indicated by a dashed line. Raw data shown as open small shapes behind means and CIs. Model information for 2019 North: GLMM (negative binomial), $X^2 = 14$, $df = 5$, $P = 0.016$, $N = 36$.

management treatment, with the greatest effect in the NPM plots (CC × PM: $\chi^2 = 6.7$, $df = 2$, $P = 0.04$).

## Weed-seed predators (ants and herbivorous carabids) in June and September

Across the experiment, we collected and identified 17 species of granivorous carabids (Fig. S6). The weed-seed predator community was dominated by *Harpalus pennsylvanicus* (1,331 individuals, 81% of all granivorous carabids in both collection periods combined). We collected this species throughout the experiment, particularly in August and September (92% of all seed predator carabids captured). Early in the season, the carabid community was more even, when other *Harpalus* species (*H. affinus, H. erraticus, H. rubripes,* and *H. faunus*), *Amara* species (*A. neoscotica* and *A. aenea*) and *Anisodactylus* species (*A. carbonaris, A. rusticus,* and *A. sanctaecrucis*) were more common. One common granivorous species, *Notiobia sayi,* was only captured in August and September (Fig. S6).

Activity-density of granivorous carabids was higher in August/September than June ($\chi^2$ = 24, $df = 1$, $P < 0.001$, Table 2), although the strength of this effect depended on the year

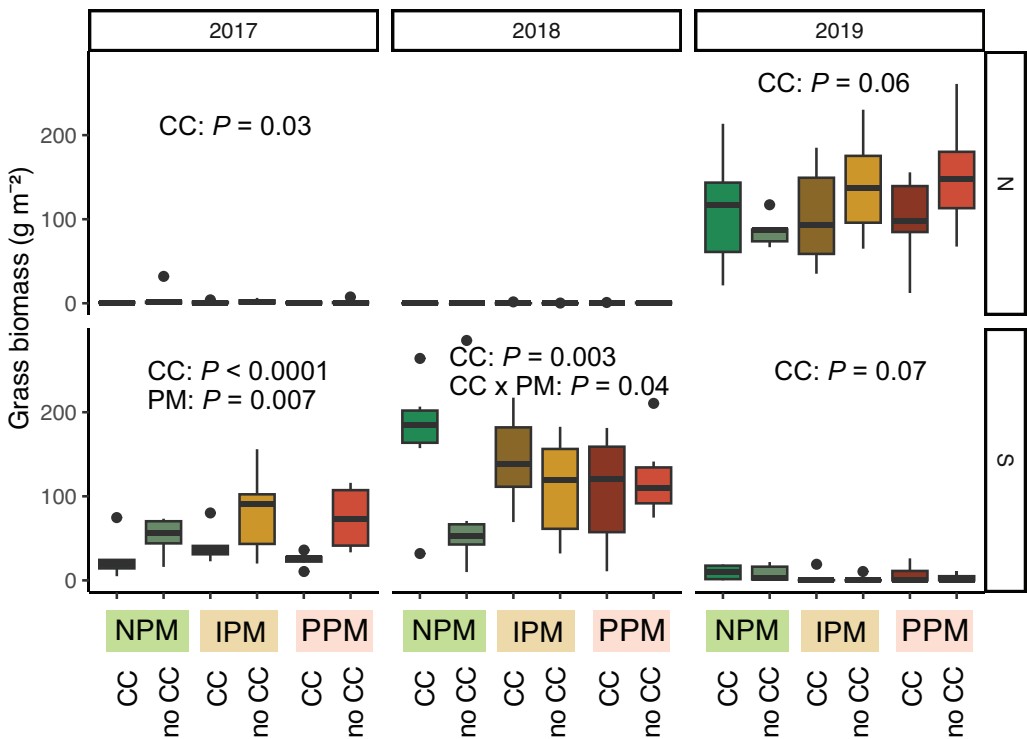

**Figure 5** **Grass biomass in August.** Boxplots of grass biomass (g) in August for in each field and year. Analyses for each field year were calculated separately. GLMM Models: 2017 North field: $X^2 = 6.3$, $df = 4$, $P = 0.17$, $N = 36$, 2017 South: $X^2 = 30$, $df = 4$, $P < 0.0001$, $N = 36$, 2018 North: $X^2 = 8.1$, $df = 6$, $P = 0.23$, $N = 36$, 2019 South: $X^2 = 5.1$, $df = 4$, $P = 0.28$, $N = 36$, 2019 North: $X^2 = 12.7$, $df = 4$, $P = 0.01$, $N = 36$. Significance of treatments included in panels where GLMM indicated cover crops (CC) or pest management (PM) had a significant effect on grass biomass.

and field (Season × Field × Year: $\chi^2 = 15.01$, $df = 2$, $P = 0.001$; Figs. 6A, 6B). Overall, activity-density of granivorous carabids was higher in non-cover cropped plots than those planted with a cover crop ($\chi^2 = 9.9$, $df = 1$, $P = 0.002$; Figs. 6C, 6D). We detected no effect of insecticide use on granivorous carabids ($\chi^2 = 0.4$, $df = 1$, $P = 0.8$).

The other granivorous group we caught in pitfall traps, ants, were affected by neither the presence of a cover crop ($\chi^2 = 0.27$, $df = 1$, $P = 0.60$) nor insecticides ($\chi^2 = 2.2$, $df = 2$, $P = 0.34$), although ants were more abundant in June than in August/September ($\chi^2 = 117.3$, $df = 1$, $P < 0.001$, Table 2), their abundance varied by year and field as well (Year × Field: $\chi^2 = 104.9$, $df = 2$, $P < 0.001$, Fig. S7).

## Weed-seed predation

To understand the potential of weed-seed predators to control weed seeds, we measured predation of two species of weed seeds (pigweed and foxtail) at three time points throughout each season. Total weed-seed predation rate depended on season and year and was affected intermittently by the presence of a cover crop and by insecticides (PM × CC × Year × Field × Season: $\chi^2 = 38.2$, $df = 8$, $P < 0.0001$, Fig. 7, Table 2). In five of six field-years, weed-seed predation increased over the season (Fig. 7). The effect

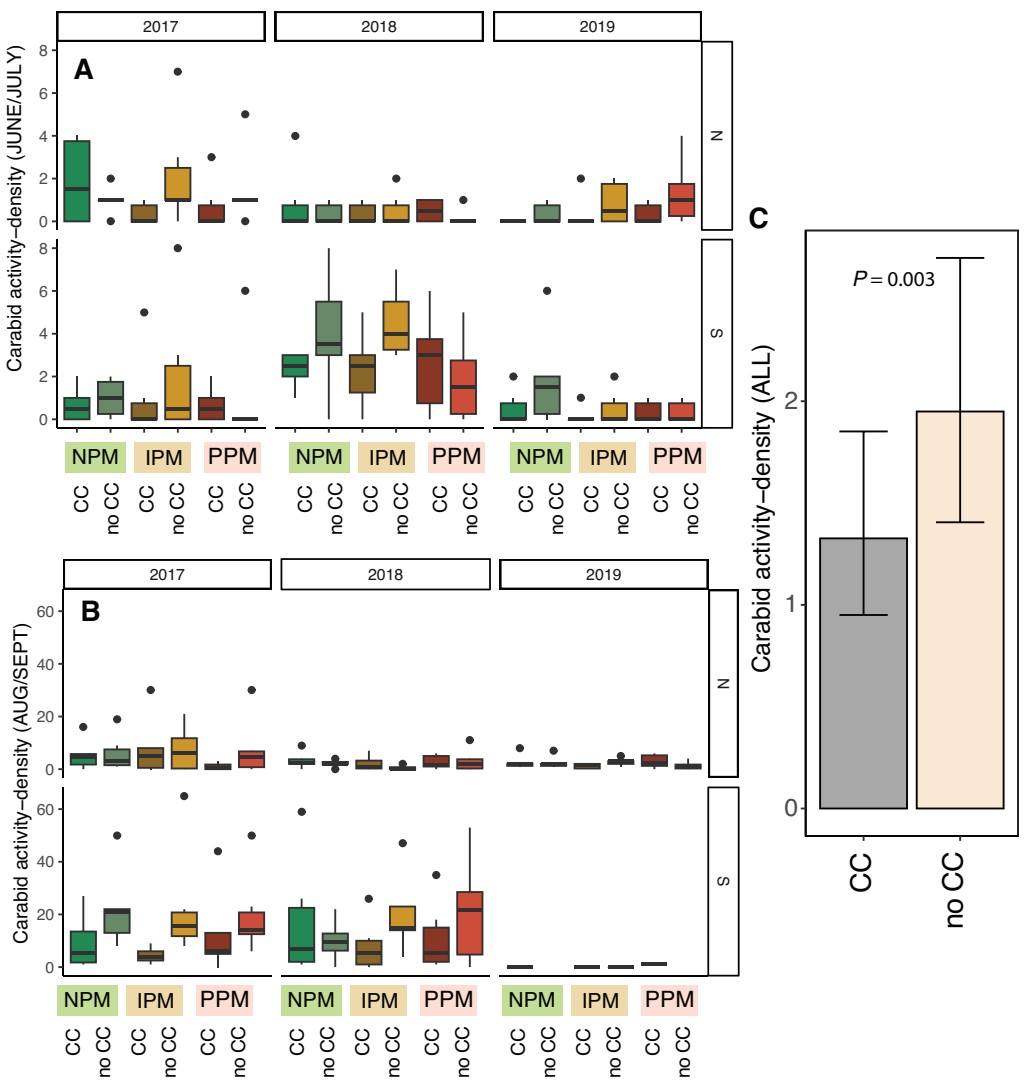

**Figure 6 Carabid activity-density from pitfall traps.** Boxplots of granivorous carabid activity-density in (A) June/July and (B) Aug/Sept for each field and year and (C) estimated marginal means ± 95% CIs across all years and fields. Model information: GLMM (Poisson distribution), $X^2 = 217$, $df = 8$, $P < 0.001$, $N = 216$. Significance of cover crop treatments (CC) included in panel C where GLM indicated cover crops had a significant effect overall (combined across all dates).

of cover crops and pest management treatment was inconsistent among years and seasons and between fields.

## DISCUSSION

The majority of corn and soybean fields in the U.S. are planted with seeds that have been treated with pesticides that typically include one or more fungicides and a neonicotinoid insecticide (*Douglas & Tooker, 2015*; *Douglas et al., 2020*). Based on previous research (*Smith et al., 2016*), we hypothesized that use of preventative fungicidal and insecticidal

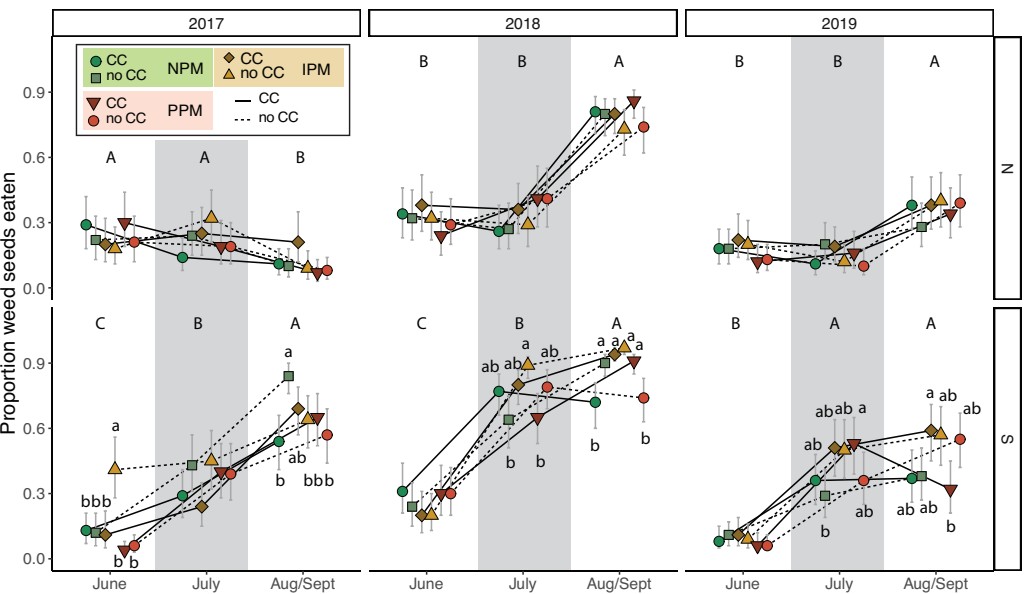

**Figure 7  Weed seed predation probability.** Estimated marginal mean probability (±95% CIs) of weed seeds eaten at each sampling event. Means for treatments with cover crops is indicated with a solid line, means without cover crop is indicated by a dashed line. Groups of means that share capitalized letters are significantly different among sampling points within a given site-year ((Tukey HSD, α = 0.0).

seed treatments would release weeds from biological control by weed-seed-infecting fungi and insect weed-seed predators. Other work suggests that even in an IPM framework, last-resort insecticide applications can have significant effects on weed-seed predation (*DiTommaso et al., 2014*), so we compared a preventive, seed-treatment-based program (PPM) and an IPM program. The data that we present here suggest that fungicides and insecticides used as seed treatments or with an IPM framework may alter weed communities, although effects are small and variable.

First, despite random assignment of treatments, at the start of the experiment the IPM plots had lower weed-seed species richness (Fig. S2). This difference does not appear to have affected forb biomass August 2017, but it may have allowed grasses to dominate in August 2017, especially where corn was growing in the South field (Fig. 5). Such differences in the weed-seed bank at the start of the experiment may have obscured treatment differences over time. However, we do not detect strong patterns year to year within plots for the weed-seed bank, suggesting that we can trust inferences from 2019 data for the weed-seed bank regardless of the patterns in 2017 and 2018 (Fig. S2).

The most compelling results of this experiment were related to *E. canadensis*, which can present a serious challenge, particularly in soybeans and no-till systems (*Klodd et al., 2017*). A single plant can produce 200,000 seeds that are wind-borne and can germinate quickly (*Molin, Parys & Beck, 2020*). Despite using glyphosate to manage weeds one week after planting, we found *E. canadensis* biomass increased from 2017 to 2019 in the North field. Glyphosate-resistant *E. canadensis* is wide-spread in Pennsylvania (*Klodd et al., 2017*), and an adjacent experiment comparing effects of herbicides on *E. canadensis* also observed

high populations of this weed following glyphosate application, leading us to suspect that this population of *E. canadensis* was resistant to glyphosate. Weed management in annual row crops has become increasingly dependent on preventative strategies, like the one we used, involving herbicide-resistant crops, and associated herbicides. Prior to the late 1990s, weeds in corn and soybean had to be controlled through tillage and a variety of selective and broad-spectrum herbicides, but since then, commercialization and rapid adoption of transgenic glyphosate-resistant corn and soybean has increased use of glyphosate (a non-selective herbicide) for post-emergence weed control. Predictably, the subsequent near-exclusive reliance on glyphosate as a weed-control strategy has resulted in evolution of weeds that are glyphosate-resistant (*Mortensen et al., 2012*). Across the major commodity-producing regions of the U.S. and elsewhere where these crops are grown, this problem has now reached epidemic levels, and for many farmers has resulted in higher crop production costs and reduced farm profitability (*Asmus, Clay & Ren, 2013*; *Sosnoskie & Culpepper, 2014*; *Evans et al., 2016*). While glyphosate resistance may be one explanation of higher *E. canadensis* biomass, taller (>10 cm) plants are also less susceptible to glyphosate, and fall seedlings be more difficult to control using glyphosate in the spring (*Klodd et al., 2017*). Thus, while we suspect glyphosate resistance, the escape of these weeds from control may be because some larger plants were able to overwinter and tolerate spring-applied glyphosate. Photos from each plot of plant cover taken just before planting, however, show very little *E. canadensis*, and no plants larger than 10 cm (see *Rowen et al., 2022* for details of imaging). Regardless if *E. canadensis* was truly resistant or of a size to be tolerant to glyphosate, alone.

Given the increasing problem of herbicide-resistant weeds, it is particularly intriguing that we found higher biomass of likely glyphosate-resistant *E. canadensis* in insecticide-treated plots (both PPM and IPM) without a cover crop (Fig. 4). The suppressive effects of cover crops on *E. canadensis* are well documented, as the cover crops can shade early season weeds, reducing their early growth (*Klodd et al., 2017*). The effect of insecticide is interesting, and we have several hypotheses that might explain higher abundance of weeds with insecticide use.

Our first hypothesis is that *E. canadensis* seeds may have landed in these plots out of random chance. We have, however, no evidence of "hotspots" of *E. canadensis* that we would expect if we had strong directional movement of seeds into plots, and the high biomasses that we measured were distributed well across the experimental plots (autocorrelation tests).

Our second hypothesis is that *E. canadensis* seeds may have been able to establish more effectively in plots with small weed-seed banks where *E. canadensis* could have established and grown vegetatively in fall and spring with less weed competition. This scenario would suggest that the decreasing weed-seed bank during the experiment (Figs. 2 and 3) favored *E. canadensis*, particularly in the North field where there was a nearby source population. Our data, however, does not support this hypothesis because we found greater biomass of *E. canadensis* in insecticide-treated plots in 2019 compared to other treatments while we did not observe an effect of pest management treatment on the weed-seed bank after 2017.

Our third hypothesis is that the insecticides decreased leaf or root herbivory of *E. canadensis*, inadvertently protecting these weeds from biological control. While we did not sample weeds for herbivores, work in the southern United States indicates that several herbivore guilds feed on *Erigeron* spp. (*Wiggins, 2021*), and it is possible that specialist or generalist herbivores were directly or indirectly decreased by insecticides and damaged *E. canadensis* plants more when unprotected by insecticides. Future work may explore if insecticides use in these field crops systems reduce herbivory on weeds.

Our fourth hypothesis is that the insecticides interfered with biological control of weed seeds by seed predators and fungal decomposers. Because we found similar effects on *E. canadensis* in both the IPM and PPM treatments, the two treatments that received insecticides, our results suggests that use of insecticides, rather than fungicides, interfered with biological control of weed seeds by insects. Based on our pitfall captures, however, we did not detect an effect of insecticidal seed treatments on activity-density of weed-seed predators (Fig. 6, Fig. S6). While we did detect variation in seed predation due to the combination of seed-applied pesticides and a cover crop (Fig. 7), this effect was inconsistent between fields and among years. Other field experiments have found that our dominant taxa of seed predators (*i.e.,* carabids and ants) can be directly and indirectly affected by neonicotinoid seed treatments (*Mullin et al., 2005*; *Douglas, Rohr & Tooker, 2015*; *Schläppi et al., 2020*). However, because pitfall traps measure activity-density, our "signal-to-noise" ratio may have been too high to detect season-long effects of weed-seed predators on the weed community. Although overall carabid activity-densities tracked with weed-seed predation (greater towards the end of the season), we did not trap frequently enough to capture day-to-day or week-to-week nuances. Carabid foraging, for example, can be influenced by weather, plant cover, and even moonlight, and thus be highly variable over short sampling periods (*Niemelä, Spence & Spence, 1992*; *Blubaugh, Widick & Kaplan, 2017*), and different carabid species may forage for different weed species and not have homogeneous responses to weeds and cover crops (*Charalabidis et al., 2019*; *De Heij & Willenborg, 2020*; *Ali et al., 2022*). While we saw similar effects in the IPM and PPM treatments, other results from this same experiment revealed that the fungal community attacking weed seeds was significantly less diverse in plots planted with seed-applied pesticides (*Palmer, 2020*). This result suggests that the fungicidal portion of seed treatments can alter soil fungal communities, possibly releasing seeds of some weed species from their pathogens (*Smith et al., 2016*). This evidence suggests a robust natural-enemy community (*i.e.,* decomposing fungi and granivorous insects) may play a disproportional role in managing weeds that are difficult to control using herbicides compared to weeds that remain tractable using chemical control practices.

We hypothesized that cover crops could ameliorate potential negative effects for weed management of insecticide and fungicide use. We found that cover crops reduced the weed-seed bank for forbs in 2018 and 2019 by 20% (Fig. 2B) but had no effect on grass after 2017 (Fig. 3B). In 2017, it is unclear why cover crops would increase rather than decrease germinable grass seed, particularly in the South field were weed seeds were particularly abundant in the weed seed bank, except perhaps due to initial differences in the weed seed bank in these plots. This effect from cover crops on weeds, however, was

only detectable early in the season. By August, weed biomass of forbs was equal in plots with and without cover crops, and the effect of cover crops on grasses was inconsistent (Fig. 5).

Unexpectedly, we also found that presence of cover-crop residue consistently decreased activity-density of carabid beetles (largely *Harpalus pensylvanicus*; Fig. 7). This contrasts with previous work that has found positive (*O'Neal et al., 2005*; *Brevault et al., 2007*; *Ward et al., 2011*; *Saenz-Romo et al., 2019*) or neutral (*Carmona & Landis, 1999*) effects of cover crops on carabids. Our data suggest that cover crops, at least at the density we planted, negatively affected activity-density of weed-seed predators, both early in the season and later in the season when carabids may not move among plots as easily (*Wallin & Ekbom, 1988*). Because they increase habitat complexity, cover crop residue, especially in the first half of the growing season, and weed biomass may slow carabid movement and reduce pitfall trap capture (*Greenslade, 1964*; *Boetzl et al., 2018*). While cover crops may reduce capture of weed-seed predators in pitfall traps (South field in Fig. 6), we did not detect consistent decreases in weed-seed predation in cover-crop plots, indicating that activity of weed-seed predators may be high enough in plots both with and without cover crops to provide sufficient weed biological control. We also previously found that carabids and ants both responded positively to plant cover present prior to planting, regardless of whether it was from a planted cover crop or from weeds (*Rowen et al., 2022*), further emphasizing that vegetation present in fields prior to planting can strongly influence beneficial arthropod populations (*Schipanski et al., 2014*).

## CONCLUSIONS

Our three-year experiment investigating impacts of preventative and integrated insect pest management on weed communities in corn and soybean fields, and potential mitigating effects of cover crops, provides insights into integrated pest and weed management. Our findings suggest that using an insecticide, either as a preventative seed treatment or in response to pest pressure, may result in small alterations in weed communities. The emergence of glyphosate-resistant *E. canadensis* in insecticide-treated plots underscores the importance of maintaining a robust natural-enemy community for effective weed management and highlights the need for even longer field experiments to detect effects of pesticides on ecosystem processes. Surprisingly, cover crops, while reducing the forb weed-seed bank, increased grass weed biomass and grass seed abundance in the weed-seed bank and had unexpected negative impacts on the activity-density of carabid beetles. These results emphasize the complexity of interactions within agroecosystems and accentuates need for holistic approaches to weed management that consider broader ecological implications of pest-control strategies.

## ACKNOWLEDGEMENTS

We thank Andrew Aschwanden, Hayden Bock, Lewis Hahn, Jennifer Halterman, Kyra Hoerr, Julie Golinski, Ken Kim, Sonia Klein, Ken Koepplinger, Roman Nakielny, Garrett

Reiter, Amanda Seow, Dan Wisniewski for assistance in the field and lab, and Austin Kirt and Corey Dillon for farm management.

### Funding

This work was funded by USDA AFRI competitive grants 2017-67013-26258 and 2017-67013-26594, a USDA NIFA pre-doctoral fellowship to E.K.R. 2018-67011-28012, and the College of Agricultural Sciences at Penn State via the National Institute of Food and Agriculture and Hatch Appropriations under Project #PEN04606 and Accession #1009362. The funders had no role in study design, data collection and analysis, decision to publish, or preparation of the manuscript.

### Grant Disclosures

The following grant information was disclosed by the authors:
USDA AFRI competitive grants: 2017-67013-26258, 2017-67013-26594.
a USDA NIFA pre-doctoral fellowship to E.K.R. 2018-67011-28012.
The College of Agricultural Sciences at Penn State via the National Institute of Food and Agriculture and Hatch Appropriations: #PEN04606, #1009362.

### Competing Interests

The authors declare there are no competing interests.

### Author Contributions

- Elizabeth K. Rowen performed the experiments, analyzed the data, prepared figures and/or tables, authored or reviewed drafts of the article, and approved the final draft.
- Kirsten Ann Pearsons performed the experiments, analyzed the data, authored or reviewed drafts of the article, and approved the final draft.
- Richard G Smith conceived and designed the experiments, authored or reviewed drafts of the article, and approved the final draft.
- Kyle Wickings conceived and designed the experiments, authored or reviewed drafts of the article, and approved the final draft.
- John F. Tooker conceived and designed the experiments, authored or reviewed drafts of the article, and approved the final draft.

### Data Availability

The data and code are available in the Supplemental Files.

### Supplemental Information

Supplemental information for this article can be found online at http://dx.doi.org/10.7717/peerj.18597#supplemental-information.

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
