# Peer review of "Insecticides may facilitate the escape of weeds from biological control"

_PeerJ, doi:10.7717/peerj.18597_

## Round 0.1 · original submission · Minor Revisions

Biological control offers a promising alternative for managing weeds without relying on herbicides, thus minimizing harmful impacts on the ecosystem. Given the potential of this approach, any new information that contributes to effective weed control is invaluable, serving as a foundation for future research in this field. However, it is essential to address specific technical details to enhance the quality of your article. I recommend thoroughly reviewing the reviewers' suggestions, carefully evaluating each one. If you disagree with any particular suggestion, it is crucial to provide clear, well-reasoned justifications to support your perspective.

·

Basic reporting

Clear, detailed. Results are relevant to the hypotheses.

Experimental design

Experimental design was good. It is unfortunate that there were very localized effects within the experiment that results in some plots being significantly different twin to effects other than the experimental treatments. But probably unavoidable?

Validity of the findings

Clearly analyzed and presented. The effects were quite small, however.

Additional comments

Annotation Summary of peerj-reviewing-102282-v0.pdf. (attached)

Note [page 5]: Maybe a different title? Insecticides cause in changes in weed assemblages as a result of impacts on biocontrol agents?

Note [page 6]: Agricultural chem/pesticide companies? Ag companies can be all kinds of stuff.

Strikeout [page 6]: And of course, agricultural companies market their products to growers emphasizing that the pesticides provide a sort of insurance policy against difficult-to-scout early season pests (Hurley & Mitchell, 2017).

Note [page 6]: Is this suggesting that farmers in cooler climates know less about this? I think this means to say that in cooler climes, proactive pesticide treatments are often not justified, but treated seed is sold to uninformed farmers irrespective?

Note [page 6]: does this refer to release of weed seeds?

Note [page 10]: keys published by?

Note [page 10]: why no spaces before citations? is this a PeerJ format requirement?

Note [page 10]: 15 what is this?1

Note [page 10]: Petri

Note [page 12]: grammar issue? consistency?

Note [page 31]: Are those 95%CIs? Surprising that the effect was significant, given the overlaps in means and CIs.

Reviewer 2 ·

Basic reporting

Specific comments:

L64: (e.g 8-11) Some cited references appear to have been formatted incorrectly. What are these references?

L85 you should rewrite this sentence to a more accurate one (releasing weed seeds from biocontrol can be confusing). It is the decrease in the effectiveness of biocontrol that leads to an increase in the number of seeds because they are less consumed.

L154: what “15” refer to?

L158: What about 2019, IPM and NPM are also the same? I guess that as IPM received treatment in 2018, then in 2019 IPM and NPM plots do not have the same history and therefore they are different… Did the experimental fields have received treatment the years before the experiments?

L160: why did you choose a different cover crop in 2018?

L163-167: Do you think that herbicide could have an impact on seed predators? Did you use herbicide only for plots with cover crops or both? Did you use the herbicide for each treatment?
Why didn’t you use herbicide a second time in 2019?

L167: (Table S1 also in 15) What “15” refer to?

L215: What 15 refer to?

L218-223: You mentioned that the number was 30 and 20 but when looking at raw data there are several occurrences of more than 30 and 20 seeds counted at the end. This is not enough rigorous because we can oppositely expect the same, so if you count 28 instead of 30, I’m not sure if 2 seeds were eaten or if it was the number of seeds at the beginning…

L227: You could also use these vertebrate exclusion cages to avoid destruction traps for assessing the weed-seed predator community…

L260: You need to check your results and P-values. When looking at the supplementary materials, the df should be equal to 1 instead of 2.

L262: I found the reduction of weed biomass due to cover crops only in the North field in 2019, please correct. It is the result of total plant biomass in spring 2019 that is similar for both fields.

L263: When looking at the supplementary materials, the df should be equal to 2 instead of 1.

L268: Unfortunately, I can’t open the supplementary material: R code and output for weed seed bank analysis. Could you please solve this issue?

L290: Captions are missing on the Fig. 3B.

L305: Unfortunately, I can’t open the supplementary material: R code and output for biomass of weeds in August data analysis. Could you please solve this issue?

L318-319: Please specify whether these results are from all treatments combined in the North field.

L354: For ants not affected by insecticides, the df should be equal to 2 instead of 1.

L363: The Statistical results for weed seed predation (in supplementary material), give a df equal to 8 instead of 4.

L363: If I understand correctly, there are 2 fields (North and South), each divided into 36 plots (6 treatments replicated 6 times), so I don’t know what five of six “site-years” refer to. I guess weed-seed predation increased over the season, except in the North field in 2017.

L378: Please, explain why you think this difference appears to have affected grass biomass but not the forb biomass in 2017. What supports this statement?

L385-387: This may highlight the importance of biological control, but it primarily indicates the problem of glyphosate resistance.

L428: How do you explain that cover crops increase the grass abundance in the weed seed bank?

L439-440: The plant density in the plot determines the activity-density of carabids. If weeds are likely to be growing faster in plots without cover crops early in the season, then it should also affect negatively the activity-density of weed seed predators. As you said for later in the season when carabids may not move among plots as easily. So how do you explain this result for early in the season?

L535: This reference is not cited in the manuscript.

L556: This reference is not cited in the manuscript.

L558: This reference is not cited in the manuscript.

L577: This reference is not cited in the manuscript.

Experimental design

Could you please provide more details on the difference between treatments (for example did you use herbicide in the NPM with a cover crop, before planting the cash crop?), and about the field history, at least in 2016 (which treatment has been used in the past)?

The raw data shows a non-equivalent number of seeds at the beginning of the experiment, and this does not seem rigorous enough to assess the correct seed predation rate. Could you please demonstrate that this is just an insignificant error in the data?

Validity of the findings

General comments:

The manuscript aims to highlight the risk of insecticide use (IPM and PPM) that could impact the community of natural enemies and therefore the effectiveness of biological control of weeds.

This field study provides interesting results regarding the use of a cover crop in terms of weed reduction before planting. However, despite the numbers of data collected, the results of weed biomass (in spring and in August) and weed-seed bank, are often inconsistent between years and treatments and it is therefore difficult to conclude.

Unfortunately, the results of granivorous insect activity-density, and weed-seed predation are too weak and indicate no effect of insecticide use. Thus, the data do not support the title, and the authors should replace it with a more appropriate title.

The main issue is that the most compelling result, which is the increase of a glyphosate-resistant weed (E. canadensis) in IPM and PPM without cover crop, is not correlated with a decrease in the effectiveness of biological control. On the contrary, you demonstrate that the activity-density of granivorous carabids was higher in non-cover cropped plots than in those planted with a cover crop.

I suggest that you continue experiments on weed seed predators and/or other arthropods (such as Hemiptera), to demonstrate whether yes or no IPM and PPM may have a significant impact on the biological control of weeds by their natural enemies.

·

Basic reporting

In general, the paper is well written and well researched, citing recent relevant literature. There are some minor type-editing issues detailed in the line by line comments below, but these are superficial. The communication style makes the intro, methods, and discussion easy to read. However, the results are more challenging due primarily to how many different types of information the authors report. Further, I am struggling to interpret meaningful information from the tables and figures that are presented in the main text. I believe more summary data information provided along with the information already in the tables, would create more context for the scale and scope of data collected. This in turn can help give more credence to the conclusions the authors draw by demonstrating that the dataset is robust.

Experimental design

The experimental randomized complete block design is well thought out and well described. The authors do a good job framing the increasing use and potential issues for preventative pest management, describing what questions they sought to answer, and the results they hypothesized would occur.

Validity of the findings

I do however have some reservations with the statistical analyses, which I believe have more to do with some information just not yet reported. Primarily, the modeling should include reported residual degrees of freedom along with the treatment degrees of freedom, so a reader knows the extent of the sample size used in these analyses.
Most importantly, I am not yet convinced of the authors’ main conclusion that seed insecticide treatment is causing an increase in Erigeron canadensis biomass. The seeds of this plant are wind dispersed and do not persist for long in seed banks. The suggested management strategy is often tillage, which appears to not be happening in these plots after the experiment began. If it is, it should be mentioned in the methods. Additionally, the authors state that the E. canadensis is herbicide resistant without detailing how this was determined. I believe the late onset of this common, noxious weed may point to a more broad introduction of the weed into the surrounding area, rather than a result of the authors’ study. If the authors believe otherwise, they should provide evidence to demonstrate to the reader how they are coming to that conclusion. Testing for spatial autocorrelation and providing additional evidence/literature would be good places to start.
Additionally, some of the other findings by the authors, namely the decrease in seed predator activity and reduction in forb biomass by various treatment combinations, have perhaps a more direct link to their experimental design. These both also impact their original questions and merit more development in the discussion.
I have some detailed suggestions in line-by-line comments below

Additional comments

The finding, if sufficiently supported, that insecticide use appears to have a link to herbicide-resistant weeds, is an important and timely one relevant to all researchers/practitioners interested in pest management, agriculture, or herbicide resistance. Further, demonstrating the efficacy of planting cover crops at reducing this herbicide-resistant weed effect provides a ready-to-implement solution, increasing the benefits of publishing this work.
With some polish and additional detail, I think this will be a valuable contribution to our knowledge to these complex systems and their management strategies
Background: first sentence appears a word is missing. “are common corn and soybean” treatments?
Line 45: First sentence is missing a word
Lines 54-56: This statement is disjointed with the rest of the paragraph. Consider removing or more explicitly linking these ideas together for the reader
Lines 64-65: Looks like these citations did not get changed to the journal style
Lines 65-74: What about non-target effects on other beneficials like pollinators? They are likely also affected, eg:
Ward, L.T., Hladik, M.L., Guzman, A., Bautista, A. and Mills, N.J., 2023. Neonicotinoid sunflower seed treatment, while not detected in pollen and nectar, still impacts wild bees and crop yield. Agrochemicals, 2(2), pp.279-295.
Main, A.R., Webb, E.B., Goyne, K.W., Abney, R. and Mengel, D., 2021. Impacts of neonicotinoid seed treatments on the wild bee community in agricultural field margins. Science of the Total Environment, 786, p.147299.
Line 87-89: Excellent point on the importance of biological control for weed management!
Line 154: another citation missing its reformat
Line 215: assuming 15 is another manuscript by the authors?
Line 234-253: In general, the modeling approaches detailed appear very sound. One concern is whether the authors checked for collinearity in their model parameters? A variance inflation factor (VIF) check can quickly investigate if too many model terms were included/terms were collinear, removing terms above an appropriate VIF threshold (2 to 10 depending on your preferred strictness). VIF can be checked quickly using the VIF function from the car package. As the authors have already done, you can still argue for including terms even if they are above this threshold should they be meaningful/biologically relevant. However, doing such a check will strengthen the authors’ ability to rely on the models.
Results:
In general, providing some summary details of the data collected will help a reader put your statistical results in context. What was the overall weed biomass measured in each treatment combination? What was the species richness in each? How many seedlings/insects were encountered? These types of information would be a meaningful addition to Tables 1 or 2, which currently do not provide much substantial information other than sampling dates. What proportion of weed biomass was forbs vs. grass? Highlighting these summary details will then make the differences/changes you observed more meaningful.
256-267: When reporting your statistical results, the degrees of freedoms are reported for the treatment, but not for the residuals, throughout. Both need to be reported so a reader can understand the sample sized used for each of these models
Lines 264-267: Why include cover-crop biomass in this? You would expect total plant biomass to be higher when a cover-crop was intentionally planted. The results section does feel a little data-dump heavy with some results being reported merely because they were measured/analyzed. Consider reducing the things you report to the details which are relevant to your research question and overall narrative to aid the reader in following your narrative.
Line 283: This sentence appears to have lost a word after consistent
Line 307-312: How exactly did you examine community composition? Do you mean you performed an analysis like an ordination or indicator analysis? Or are you using an encounter abundance threshold to determine what species counted as characterizing a community? Either approach is reasonable, but you need to describe how you are determining this.
Lines 382-383: Please detail how it was confirmed that this was glyphosate-resistant E. canadensis. Was it actually tested or assumed?
Erigeron is also wind-dispersed. You point out a spatial occurrence of E. canadensis emerging in the North field in the final year of the study. Are you sure this result is related to your experiment, and not an edge effect resulting from increasing Erigeron source populations outside of the experimental plots? E. canadensis does not persist long in a seed back, so it’s unlikely the large increase in 2019 was from dormant seed. A Moran’s I test for spatial autocorrelation on sample location seed bank diversity/abundance could help you make the argument that your observed increase was a treatment rather than a wind-dispersed edge effect. As the authors point out, these areas used to be tilled, and tillage is actually one of the common effective treatments for E. canadensis so it may actually be an artifact of the reduction in tillage practices, which is worth exploring in the discussion.
Figure 1: It looks like the y-axis for these plots is large for a single outlier in the 2017 PPM treatment in the South field. I’d suggest limiting the axis to around 125-150 so that the data are more easily seen and distinguishable at the loss of the outlier
Figure 2: Caption should read “Forb abundance…”? What is the axis value for the bars in panel B? It appears that there is a trend of abundance decreasing in general each year regardless of treatment combination. What do you think caused this?
Figure 3: Again there appears to be a decrease in abundance over time. Perhaps this more open seedbank enabled E. canadensis to more successfully introduce into the plots? Figure 4 shows a minor E. canadensis intro in 2018 before the major one in 2019
Figure 5: does this include or exclude E. canadensis?
Figure 6: What are the error bars for panels B and D? there seems like a large amount of overlap for things that are significantly different

---

## Round 0.2 · Minor Revisions

I appreciate your positive and constructive attitude toward the suggestions of reviewers. However, your article needs some minor revisions to improve before publishing. Please carefully read the reviewer's comments and consider each of them. If you find yourself in disagreement with any particular suggestion, it would be beneficial to provide clear and well-reasoned justifications for your perspective.

·

Basic reporting

This is a review of a revised manuscript. The authors have addressed all concerns adequately.

Experimental design

Good. unfortunately some variation within the experimental layout may have ,asked actual experimental effects.

Validity of the findings

I think the finding are valid given the experimental constraints.

Additional comments

revised manuscript, comments have been addressed.

·

Basic reporting

Authors made the attempt to incorporate reviewer feedback for basic reporting critiques and the language and style of their writing remains excellent. No further improvements on the basic reporting are needed

Experimental design

remains decent, no further comments

Validity of the findings

In general, the authors responded to reviewer comments in their response to the reviewers, but did not address the overall reviewer concerns/comments within the text of the manuscript itself. If a reviewer is bringing up an additional point/concern, it is likely that other readers will also have that concern, and it is important that these comments receive acknowledgement and/or consideration by the authors so that the overall length to which the authors findings can reach is put in appropriate context. Detailed continued concerns below.

Additional comments

Lines 48-51: if the ideas that the authors are trying to express are “that in cooler, more northern latitudes, pesticide seed treatments are not offering protection, and growers often either don't realize they are using them, and/or can't get seed without seed treatments, while in the southern US, they offer more economic benefit” then the authors should explicitly state those ideas. As written, these lines are vague and uninformative to a reader who has not already reached the same conclusions.

Lines 231-252: The authors still need to address the potential for collinearity in their model terms. This is a simple test, and in the first round of revision comments, the package and function to do this test was detailed. The notion that the model terms are designed to be orthogonal does not remove the possibility that they can be collinear, and until a test is done to demonstrate that they are not, the assumptions of the models are not met and the interpretation of the model results are suspect. If the model terms do have collinearity as indicated by high VIF values, the authors need to either address this with new modeling, or discuss why they think the models remain appropriate for interpretation of their study within the text of the manuscript.

Lines 383-385: The authors have now conducted a spatial autocorrelation test, but the results are not explored anywhere in the discussion. Use your result from the Moran test to support your argument and tell the reader what you think that result means in the context that if the seeds were seed dispersed, or differentially pre-dispersed within the seed bank, you would expect them to be spatially auto-correlated.

Figures 2 and 3: These figures show a general decrease in both forbs and grass abundance over time. Responding “Yes that is certainly possible” is an inadequate response to a reviewer comment. Please address in the discussion the potential for a diminishing seedbank generally over time to have allowed establishment of other plants like the glyphosate resistant E. canadensis, and detail in the discussion what leads you to the conclusion that it is an insecticide treatment, rather than another stochastic event, that is linked to your observed trend of increasing resistant E. canadensis. The conclusion of this manuscript is intriguing, but borders on reaching beyond what its data can support as the experiment was not designed to investigate what emerged as the more intriguing finding. Putting your conclusions in the context of the other possibilities, and acknowledging that those other scenarios are possible or detailing why you think they are not likely, is an important conversation needed for the reader.

Supplemental materials: The authors chose to list the total sample size for their various models in the supplemental materials. This information is fundamental to understanding the power of the models used, critical for interpreting findings appropriately, and should be incorporated into the main text

---

## Round 0.3 · accepted · Accept

I appreciate your positive and constructive attitude toward the suggestions of reviewers. I believe your manuscript is now ready for publication. We look forward to your next article.

·

Basic reporting

The manuscript meets all the criteria of basic reporting. It is well written, conveying its ideas clearly

Experimental design

The experimental design is well laid out and implemented

Validity of the findings

The authors' exploration of the hypotheses for their findings of increased Erigeron and how those potential hypotheses relate to their additional analyses/information create an appropriate context for exploring their findings. They have also now addressed my previous concerns by providing additional information/context for their statistical analyses and figures

Additional comments

I appreciate the authors incorporation of both rounds of feedback into the manuscript and commend them for their thorough efforts. I do believe providing the VIF results in the supplemental materials is sufficient for this publication